# Increased interregional virus exchange and nucleotide diversity outline the expansion of chikungunya virus in Brazil

The emergence and reemergence of mosquito-borne diseases in Brazil such as yellow fever, zika, chikungunya, and dengue have had serious impacts on public health. Concerns have been raised due to the rapid dissemination of the chikungunya virus across the country since its first detection in 2014 in Northeast Brazil. In this work, we carried out on-site training activities in genomic surveillance in partnership with the National Network of Public Health Laboratories that have led to the generation of 422 chikungunya virus genomes from 12 Brazilian states over the past two years (2021–2022), a period that has seen more than 312 thousand chikungunya fever cases reported in the country. These genomes increased the amount of available data and allowed a more comprehensive characterization of the dispersal dynamics of the chikungunya virus East-Central-South-African lineage in Brazil. Tree branching patterns revealed the emergence and expansion of two distinct subclades. Phylogeographic analysis indicated that the northeast region has been the leading hub of virus spread towards other regions. Increased frequency of C > T transitions among the new genomes suggested that host restriction factors from the immune system such as ADAR and AID/APOBEC deaminases might be driving the genetic diversity of the chikungunya virus in Brazil.

Mosquito-borne viral diseases have impacted the lives of millions of people across several populations in the tropics and subtropics[1,2]. This scenario prompted the World Health Organization (WHO) to issue a guideline in 2017 with strategies for global vector control response aiming to reduce the burden and threat of vector-borne diseases, such as Dengue, Zika, and chikungunya fever, by 2030[3]. Five years later, considering the public health implications brought by the coronavirus disease (COVID-19) pandemic, the WHO outlined ten proposals to strengthen health emergency preparedness, response, and resilience[4]. One of the proposals calls for the development and establishment of a collaborative surveillance system with improved laboratory capacity for pathogen and genomic surveillance that would guide the public health response.

Several genomic surveillance initiatives have been carried out in the last few years to build knowledge regarding the genetic diversity and transmission dynamics of arboviruses in Brazil, such as

chikungunya virus (CHIKV)[5–12]. CHIKV infection can cause long-lasting effects such as debilitating arthritis and arthralgia, and there are no effective treatments available[13]. Vaccine candidates in development have reached phase 2 and 3 clinical trials using attenuated virus derived from the Indian Ocean lineage (IOL) or virus-like particle containing recombinant structural proteins derived from a Senegalese viral strain[14,15]. Genomic sequencing has revealed that the first case of chikungunya fever (CHIKF) reported in Brazil was an infection by the Asian lineage introduced in a northern state in 2014, while another case reported seven days later in a north-eastern state represented the first known introduction of the East-Central-South-African (ECSA) lineage in the country[12]. The establishment of the CHIKV in the Brazilian territory was followed by several outbreaks reported across the country, accounting for more than 200 thousand confirmed cases in only the last two years[16]. Viral genomic data from Brazilian cases has revealed that the ECSA lineage is widespread throughout the country[17] and has

✉ e-mail: luiz.alcantara@ioc.fiocruz.br; ana.bispo@ioc.fiocruz.br; giovanetti.marta@gmail.com

been linked to fatal cases observed in both risk and non-risk groups (young adults and people without comorbidities)[18].

Viral genomic surveillance activities have been driven by the rapid development of DNA sequencing technology and bioinformatics tools for genomic data analysis[19]. Such tools have allowed the characterization of the genome and dispersal patterns of emerging and reemerging pathogens[6,9,20]. The use of such tools during the COVID-19 pandemic allowed, for example, the rapid identification of emerging mutations likely associated with increased transmissibility and immune escape[21]. Despite technological advances and the high number of CHIKF cases reported in recent years in Brazil, the amount of genomic data available in public databases has consisted of genomes from localized outbreaks that could be limited in terms of representativeness across different states and outbreak events. By generating CHIKV sequence data, extensive genomic surveillance efforts in Brazil can provide updated information on the genetic pool of the viral population circulating in the country, fostering new studies on virus evolution and disease treatment.

Recurring outbreaks demonstrate that CHIKV is currently endemic in Brazil. The existence of abundant vectors, together with adequate climatic conditions for vector survival in areas of high population density, create conditions that can modify the adaptive landscape, allowing the continued expansion and evolutionary adaptation of CHIKV[22–24]. Faced with a scenario of limited availability of genetic information on potential strains causing a rapid increase in the number of CHIKF cases over the past two years in Brazil, in this work, we carried out on-site training activities in genomic surveillance in 12 Brazilian states covering four geographic regions to increase the number of available viral genomic sequences. This has allowed comprehensive monitoring of the expansion of the ECSA lineage and its variants circulating in different states, in addition to the characterization of the most up-to-date structured phylogeny of the CHIKV-ECSA lineage in the country.

## Results
### East, Central, and South African (ECSA) lineage monitoring through countrywide genomic surveillance

Nanopore sequencing was performed on selected CHIKV-positive samples provided by state public health laboratories from 12 states across 4 geographic regions (Northeast, Midwest, Southeast, and South) of Brazil (Fig. 1a) during the years 2021–2022, which saw a significant increase in the number of CHIKF cases reported across different Brazilian regions, with a peak incidence rate of more than 20 cases per 100,000 population, and a total of more than 312,000 suspected cases reported nationally in this 2-year period (Fig. 1b). Due to the portability and easy setup of the nanopore sequencing protocol, which allows data generation in less than 24 h, the collaborative work with the public health laboratories was able to not only generate genomic data but also promote on-site genomic surveillance training activities for the local laboratory staff. This approach combining wet lab and data analysis training allowed local teams to understand how genomic data can be linked to demographic information in order to produce comprehensive and relevant inferences regarding the epidemiology and evolution of CHIKV circulating in Brazil.

A total of 425 CHIKV-positive samples were subjected to nanopore whole-genome sequencing, with 84.94% ($n = 361$) of these samples originating in the Northeast region, consisting largely of samples collected from the state of Bahia ($n = 102$) (Fig. 1a, Table 1, and Supplementary Fig. 1). These samples presented a mean RT-qPCR cycle threshold (Ct) value of 24.04 (ranging from 11 to 35.90) (Table 2). Patients' mean age upon sample collection was similar for both females and males (39 years of age), with 57.88% ($n = 246$) of the participants identified as female (Table 2). The clinical status of patients at the time of sample collection, and travel history data were not available for these samples.

Multiplex PCR-tiling amplicon sequencing on MinION allowed the recovery of 425 genomic sequences from CHIKV with a mean genome coverage and sequencing depth of 90.98% (range 31.80 to 96.19%) and 3290x (range 58 to 129,706), respectively (Supplementary Data 1). Of the 425 sequences, 14 were recovered from old CHIKV-PCR samples collected in Bahia state during July–August 2015 and stored since. These isolates had a mean genome coverage of 70% (range 31.80 to 92.9%). The remaining genomes have an associated collection date ranging from April 2019 to June 2022. To better capture the phylogenetic signal, only the sequences with genome coverage over 60% ($n = 422$) were considered for further analysis (discarded sequences are listed in the methods).

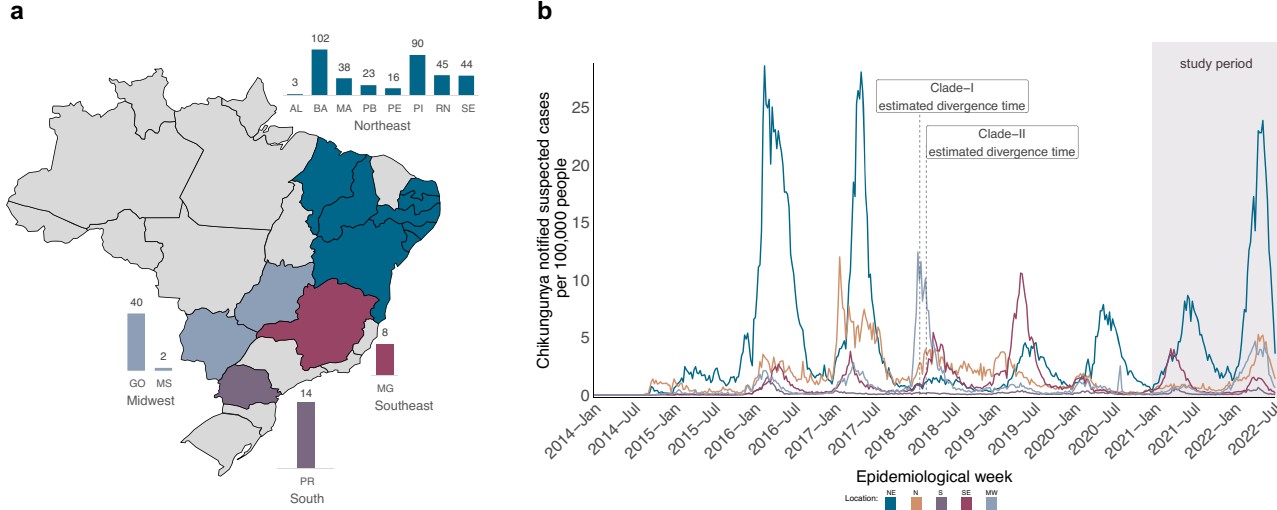

**Fig. 1 | Spatiotemporal distribution of chikungunya fever (CHIKF) in Brazil. a** Map of Brazil displays states colored according to the location where samples were collected and sequenced. Bar plots indicate the number of isolates obtained from each state. States abbreviations: AL = Alagoas, BA = Bahia, MA = Maranhão, PB = Paraíba, PE = Pernambuco, PI = Piauí, RN = Rio Grande do Norte, SE = Sergipe, MG = Minas Gerais, PR = Paraná, GO = Goiás, MS = Mato Grosso do Sul. **b** Time series of monthly reported CHIKF suspected cases normalized per 100 K individuals in five Brazilian macroregions over 2014–2022 (until epidemiological week 28). Epidemic curves are colored according to geographical macroregion: N = North, NE = Northeast, MW = Midwest, SE = Southeast, S = South. The shaded rectangle indicates the period in which samples were collected for this study. See "Data availability" for Source data.

**Table 1 | Number of chikungunya virus-positive samples sequenced during the genomic surveillance in Brazil, 2021–2022, by geographical origin**

| Region and State | # Samples (total $n = 425$) |
|---|---|
| Northeast | ($n = 361$) |
| Alagoas (AL) | 3 (0.71%) |
| Bahia (BA) | 102 (24.00%) |
| Maranhão (MA) | 38 (8.94%) |
| Paraíba (PB) | 23 (5.41%) |
| Pernambuco (PE) | 16 (3.77%) |
| Piauí (PI) | 90 (21.18%) |
| Rio Grande do Norte (RN) | 45 (10.59%) |
| Sergipe (SE) | 44 (10.35%) |
| Midwest | ($n = 42$) |
| Goiás (GO) | 40 (9.41%) |
| Mato Grosso do Sul (MS) | 2 (0.47%) |
| Southeast | |
| Minas Gerais (MG) | 8 (1.88%) |
| South | |
| Paraná (PR) | 14 (3.29%) |

**Table 2 | Demographic and laboratory characteristics of chikungunya virus-infected patients**

| | Patients ($n = 425$) |
|---|---|
| **Sex** | |
| Female | 246 (57.88%) |
| Male | 179 (42.12%) |
| **Mean age on sample collection (SD)** | |
| Female | 39.10 ± 18.83 |
| Male | 39.37 ± 22.03 |
| Median days of symptoms prior to sampling (SD) | 3.47 ± 8.52 |
| RT-qPCR mean Ct (cycle threshold) value (SD) | 24.04 ± 4.15 |

*SD* standard deviation.

All the newly recovered genomes were assembled using Genome Detective software which also classified all of them as belonging to the East, Central, and South African (ECSA) lineage. To investigate the phylogenetic relationship of the new sequences with other Brazilian and non-Brazilian sequences available in public databases, we built a global dataset ($n = 1987$) composed of 1565 CHIKV genomes retrieved from GenBank NCBI in addition to 422 sequences from this study. It is noteworthy that the two years of genomic surveillance activities of this study contributed to a more than doubling of the number of CHIKV genomes from Brazil available in the NCBI (by then there were 332 complete sequences) since the virus emerged in the country in 2014 (Fig. 2c).

**Updated CHIKV phylogeny reveals two distinct emerging subclades**

A preliminary Maximum Likelihood (ML) tree was reconstructed using the global dataset that showed all Brazilian sequences grouped in the ECSA clade (Supplementary Fig. 2). It can also be observed from the ML tree that most of the new sequences formed two well-distinct derived clades, where it can be noticed that sequences collected in different geographical regions were closely related (see ML tree in Supplementary Data 2). To investigate in more detail the phylogenetic features of these clades within a time-aware evolutionary framework, which can benefit from the increased amount of genomic data

obtained in this study, we performed a Bayesian phylogenetic analysis using a downsampled dataset ($n = 713$) mostly composed of Brazilian sequences.

Root-to-tip genetic distance regression indicated that the downsampled dataset presented sufficient temporal signal ($R^2 = 0.70$ and correlation coefficient = 0.83) to infer a time-measured phylogeny (Fig. 2b). Consistent with the ML tree, the inferred Maximum Clade Credibility (MCC) tree also revealed two distinct more derived clades (henceforth clade I and II) formed mainly by 2021–2022 sequences (Fig. 2a). The Bayesian evolutionary analysis estimated the time of the most recent common ancestor (tMRCA) of clade I to be late January 2018 (95% highest posterior density (HPD): December 2017 and March 2018), while clade II presented a slightly late tMRCA estimated to be early February 2018 (95% HPD: January 2018 and March 2018).

Some composition differences can be noticed between these clades. Clade I comprises sequences from 14 distinct states mostly collected from 2021 to 2022 ($n = 304$) and from northeastern Brazil (62.38%, $n = 204$), with sequences from Sergipe (13.5%, Northeast), Minas Gerais (1.8%), São Paulo (16.2%, Southeast), Goiás (11.6%), Mato Grosso do Sul (0.6%, Midwest), and Paraná (4.3%, South) states uniquely present in this clade (Fig. 2a). Meanwhile, clade II is mostly composed of sequences collected in 2022 from northeastern states (87.5%, $n = 147$), with a total of 8 states sampled and Piaui state being the most represented (46.42%, $n = 78$) (Fig. 2a).

A closer look at sequence distribution inside these clades reveals recurrent virus movement between states and regions, with mid-western isolates closely related to isolates from the Northeast and from the state of Minas Gerais (Southeast) in clade I. It can also be noticed in clade I that several distinct CHIKV introductions occurred in the state of Goiás (Midwest) and in northeastern states (Bahia, Rio Grande do Norte, Sergipe, Paraiba, Pernambuco, and Piaui) (Fig. 2a). Contrarily, the clade I sequence distribution reveals that apparently, only one viral introduction event has happened in the southern state of Paraná, sharing a most recent common ancestor (dated from Jun. 2019 to Jun. 2020, 95% HPD, posterior probability = 1.0) with isolates from the state of São Paulo (Southeast cluster). Similarly, clade II displays viral exchange between states, especially from the Northeast, as indicated by a single well-supported subclade (posterior = 1.0), dated from Jan. 2020 to Dec. 2020 (95% HPD) and dominated by sequences from northeastern states (Piauí, Bahia, Rio Grande do Norte, Paraíba, Pernambuco, Maranhão, and Alagoas) (Fig. 2a).

**CHIKV dispersal in Brazil has been mainly seeded by the Northeast**

In face of the recurrent virus movement observed across the country, as indicated by our MCC phylogeny, we employed a Bayesian phylogeographic approach to reconstruct the spatial dispersal dynamic of CHIKV in Brazil (closely related sequences from Haiti and Paraguay were included) and to estimate the ancestral locations of clades I and II. The resulting phylogeny kept the topology from the MCC tree shown in Fig. 2a and revealed the Northeast as a leading source of CHIKV transmission in Brazil, seeding the network of frequent virus exchange among states mainly from the Northeast, Southeast, and Midwest, as indicated by the location probability of the 5 early branching events inferred by the discrete phylogeography (Fig. 2d and Supplementary Fig. 3). Moreover, the ECSA lineage circulating in Brazil extended its transmission network by reaching other countries in the region such as Paraguay and Haiti, likely via the Midwest and Northeast of Brazil, respectively. An alternative approach, using a transmission network generated from transition states summarized from the Bayesian phylogeography and centrality metrics, also indicated the Northeast as a source (Source Hub Ratio of 0.66) in the network for the CHIKV spread in the country, where intense interactions are displayed between the Northeast and Southeast (Supplementary Fig. 4 and Supplementary Table 1). Moreover, the discrete state ancestral reconstruction

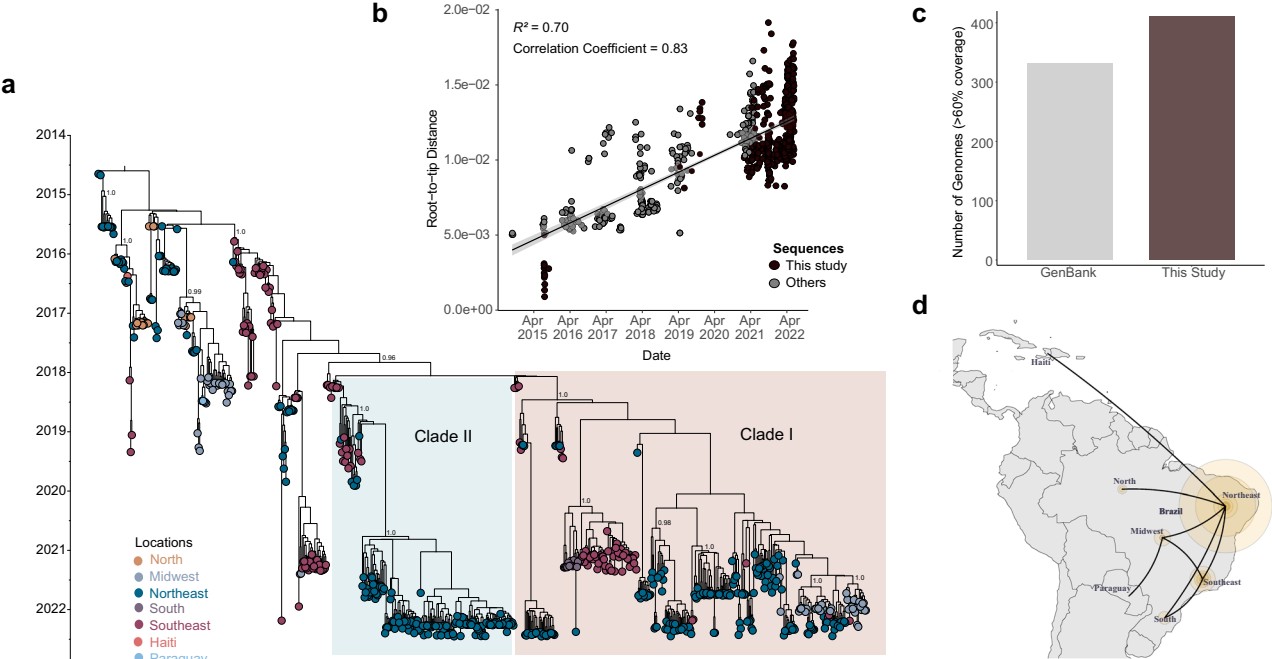

**Fig. 2 | Time-measured phylogeny of chikungunya virus ECSA lineage in Brazil.**
**a** Maximum Clade Credibility tree reconstructed using 706 sequences from Brazil (in addition to 5 sequences from Paraguay and 2 from Haiti) and a molecular clock approach. Numbers in black show clade posterior probabilities of main nodes. Some posterior probability values were omitted for clarity. Tip colors represent the sampling location. **b** Root-to-tip genetic distance regression in a maximum likelihood phylogeny of the CHIKV ECSA lineage ($n = 713$). New sequences are colored in black. The black line represents the medium values of the linear regression while

the light gray bands around the line represent the standard error. $R^2$ indicates the coefficient of determination. **c** Number of genomes generated in this study (with >60% genome coverage) compared to the number of Brazilian CHIKV ECSA sequences available on the GenBank up to 27th Jan. 2023. **d** The spatial spread of CHIKV in Brazil estimated, using SPREAD4 software, under a discrete diffusion model employed in the Bayesian Phylogeographic approach using a dataset with 471 sequences. Size of colored circles was scaled by location posterior support. See Data Availability for Source data.

employed in our Bayesian analysis estimated that both clades I and II might have emerged in the Southeast region with a location probability of 1.0 and with estimated divergence times (clade I: 95% HPD Nov. 2017–Feb. 2018; clade II: Dec. 2017–Mar. 2018) (Supplementary Fig. 3) comparable to those obtained in the MCC tree from Fig. 2a inferred using a comprehensive dataset.

These estimates place the divergence time of clades I and II in the period that marks the return of CHIKV-increased transmissions after two main epidemic seasons registered from 2016 to 2017 when a total of more than 565 thousand disease cases were notified in the country (Fig. 1b). From the time series graph of CHIKF cases, we can see an increase in the incidence rate around early 2018 for the Midwest and North regions. In that same period, clades I and II were estimated to emerge in the Southeast region, which also presented an increased incidence rate. Since then, a seasonal epidemic pattern has been observed in the CHIKV transmission dynamic with incidence peaks being displayed in the first months of the following years (Fig. 1b). Although the Northeast displayed a slightly later peak in its incidence curve in 2020 and 2021, but not in 2022, when compared to the other regions. These changes in the incidence curve patterns could be explained by differences in local climatic conditions within regions associated with differences in population susceptibility and immunity to the virus, leading to the differential spread of the virus in other parts of Brazil. Alternatively, it has been suggested that spatial heterogeneity associated to virus spread could determine CHIKV epidemics in urban environments[25]. Moreover, surveillance and reporting biases cannot be ruled out as the 2020–2021 period comprises the time when the states were heavily impacted by rapid increase in the number of SARS-CoV-2 infections, which subsided in 2022.

Consistent with the phylogeography and the transmission network analyses, the CHIKF case time-series plot presents the Northeast

region as a major source of virus transmission in the country, as shown by the consecutive incidence peaks registered for that region in the last three years.

## Transition changes outline the evolutionary expansion of the Brazilian ECSA lineage

Despite having close divergence times, clades I and II have different sequence compositions that likely explain their apparent branching distance. While sequences from clades I and II were obtained from patients with similar median age, a significant difference ($p < 0.05$) was observed between the median RT-qPCR Ct value of the samples of each clade (Fig. 3a). This difference in the median Ct values, however, might have been caused by an imbalance present in the dataset used for comparison since clade I has almost twice as many sequences compared to clade II. Alternatively, the between-clades inconsistency observed in the time between symptoms onset and sample collection could have affected sample Ct values.

We also assessed these clades as to which selective regime they are likely to be subject to. The results from the BUSTED analysis provided evidence that the envelope polyprotein coding region experienced positive diversifying selection in both clades I ($p = 1.774e-8$) and II ($p = 1.840e-11$) (Supplementary Table 2). For comparison, we employed a second method, MEME, that identified 12 sites under positive selection for clade I, whereas 13 sites were identified in clade II. Of these sites, 7 are shared by both clades, while exclusive positive selected sites were reported for each clade (Supplementary Table 3). These results suggest the active status of these clades and consequently provide evidence of the continuous evolutionary expansion of the Brazilian ECSA lineage.

Separate sequence alignments representative of each clade revealed a significant difference between the median frequency of 3

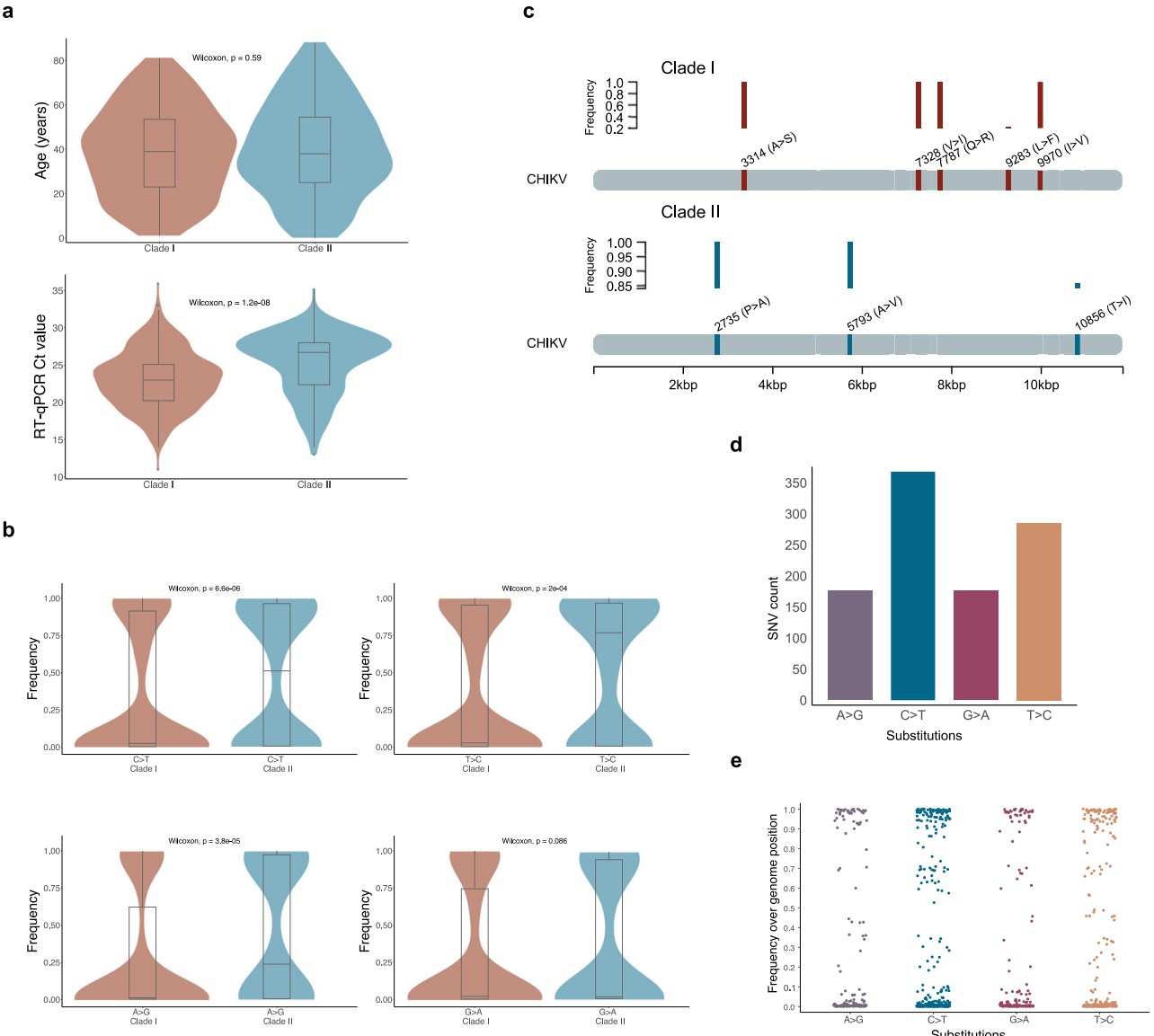

**Fig. 3 | Genetic and demographic aspects of the newly generated chikungunya virus (CHIKV) sequences. a** Violin plot and boxplot of the patient's age and samples' cycle threshold (Ct) value distributions by clades I (n = 255 biologically independent samples) and II (n = 147 biologically independent samples). Mann–Whitney U (Wilcoxon) two-sided test with a significance level alpha = 0.05. **b** Violin plots and boxplots of the frequency distributions for each class of single nucleotide variation (SNV) identified across genome positions of sequences from clades I (n = 327 biologically independent samples) and II (n = 168 biologically independent samples). Mann–Whitney U (Wilcoxon) two-sided test with a significance level alpha = 0.05. **a, b** Data visualized as violin plots (outer shape representing the data kernel density) and box-and-whisker plots (box represents the median value and 25 and 75% quartiles, the whiskers display the minimum and maximum values, 1.5x interquartile range; points, outliers). **c** Genomic positions and frequency (represented by plotted bars) of the non-synonymous substitutions uniquely identified in the sequences from clades I and II. Numbers represent the genomic position of each substitution. **d** Bar plot of the absolute number of CHIKV genome mutated positions for each class of single nucleotide variation (SNV) identified in the alignment of the new CHIKV sequences (n = 422 biologically independent samples). **e** Scatterplot of the frequency distributions for each class of single nucleotide variation (SNV) identified in the alignment of the new CHIKV sequences (n = 422 biologically independent samples). Each dot represents the frequency of a distinct mutated genome position. **b, d, e** Letters in the x-axis represent nucleotides: A−adenine, G−guanine, C−cytosine, T−thymine. See "Data availability" for Source data.

classes of single nucleotide variation (SNV) identified in the clades. Clade II presented a higher median frequency of SNV of type C > T, A > G, and T > C transitions (Fig. 3b). By comparing the mutational profile of the two clades, we identified 27 SNVs exclusively present and shared by sequences from clade II, with three of them being non-synonymous substitutions leading putatively to amino acid change such as a T288I substitution in the E1 protein (Table 3 and Fig. 3c). The other two non-synonymous substitutions were identified in two non-structural proteins (nsP2-P352A and nsP4-A43V).

Sequences from Clade I, on the other hand, presented 13 exclusive SNVs, of which six are non-synonymous mutations present in two

nonstructural proteins (nsP2 and nsP4) and three structural proteins (capsid, 6k and E2) (Table 3 and Fig. 3c). The capsid sequence from this clade presented two contiguous transition substitutions in the same codon (7787-7788) resulting in a Q74R change. Moreover, both clades contain genomes harboring the E1-T98A mutation (121/327 sequences in clade I, and 31/167 sequences in clade II), while only clade II reported 113 sequences (67%) presenting another mutation, the V264A in the E2. These two mutations have been associated with increased virus infectivity in *Aedes ssp.* when present in distinct genetic backgrounds (E1-226V with E1-T98A or E1-226A with E2-V264A)[26,27]. However, the E1-226V mutation, as well as other E1 (K211E) and E2 (D60G, R198Q,

**Table 3 | List of single nucleotide variations (SNV) uniquely identified for clades I and II**

| Genomic position[a] | Reference base | Altered base | Substitution type | aa[b] change | Clade | Genomic region | aa[c] position | Frequency[d] |
|---|---|---|---|---|---|---|---|---|
| 3208 | C | T | S | D > D | 1 | nsP2 | 509 | 1.00 |
| 3217 | T | C | S | Y > Y | 1 | nsP2 | 512 | 1.00 |
| 3277 | C | T | S | S > S | 1 | nsP2 | 532 | 1.00 |
| 3295 | G | A | S | P > P | 1 | nsP2 | 538 | 1.00 |
| **3314** | **G** | **T** | **NS** | **A > S** | **1** | **nsP2** | **544** | **1.00** |
| **7328** | **G** | **A** | **NS** | **V > I** | **1** | **nsP4** | **555** | **1.00** |
| **7787** | **A** | **G** | **NS** | **Q > R** | **1** | **capsid** | **74** | **1.00** |
| **7788** | **G** | **A** | **NS** | **Q > R** | **1** | **capsid** | **74** | **1.00** |
| 8641 | C | T | S | L > L | 1 | E2 | 34 | 0.68 |
| **9283** | **C** | **T** | **NS** | **L > F** | **1** | **E2** | **248** | **0.22** |
| 9285 | C | T | S | L > L | 1 | E2 | 248 | 0.73 |
| 9936 | T | C | S | C > C | 1 | 6K | 42 | 1.00 |
| **9970** | **A** | **G** | **NS** | **I > V** | **1** | **6K** | **54** | **1.00** |
| 1754 | T | C | S | L > L | 2 | NSP2 | 25 | 1.00 |
| 1759 | T | A | S | V > V | 2 | NSP2 | 26 | 1.00 |
| 1783 | T | G | S | R > R | 2 | NSP2 | 34 | 1.00 |
| 1789 | A | G | S | Q > Q | 2 | NSP2 | 36 | 1.00 |
| 1795 | T | C | S | L > L | 2 | NSP2 | 38 | 1.00 |
| **2735** | **C** | **G** | **NS** | **P > A** | **2** | **NSP2** | **352** | **1.00** |
| 2767 | A | G | S | G > G | 2 | NSP2 | 362 | 1.00 |
| 2887 | T | C | S | L > L | 2 | NSP2 | 402 | 1.00 |
| 3403 | T | C | S | F > F | 2 | NSP2 | 574 | 1.00 |
| 3901 | A | G | S | V > V | 2 | NSP2 | 740 | 1.00 |
| 3916 | T | C | S | F > F | 2 | NSP2 | 745 | 1.00 |
| 4711 | T | C | S | D > D | 2 | NSP3 | 212 | 1.00 |
| 4837 | A | G | S | S > S | 2 | NSP3 | 254 | 1.00 |
| 4843 | C | T | S | P > P | 2 | NSP3 | 256 | 1.00 |
| 4939 | T | C | S | S > S | 2 | NSP3 | 288 | 1.00 |
| 5783 | C | T | S | L > L | 2 | NSP4 | 40 | 1.00 |
| **5793** | **C** | **T** | **NS** | **A > V** | **2** | **NSP4** | **43** | **1.00** |
| 5807 | C | T | S | L > L | 2 | NSP4 | 48 | 1.00 |
| 5857 | T | C | S | Y > Y | 2 | NSP4 | 64 | 1.00 |
| 5921 | T | C | S | L > L | 2 | NSP4 | 86 | 1.00 |
| 5956 | C | T | S | T > T | 2 | NSP4 | 97 | 1.00 |
| 6994 | T | C | S | N > N | 2 | NSP4 | 443 | 1.00 |
| 7213 | A | G | S | T > T | 2 | NSP4 | 516 | 1.00 |
| 9153 | T | C | S | G > G | 2 | E2 | 204 | 0.89 |
| 9642 | T | A | S | T > T | 2 | E2 | 367 | 0.80 |
| **10856** | **C** | **T** | **NS** | **T > I** | **2** | **E1** | **288** | **0.86** |
| 11241 | T | C | S | G > G | 2 | E1 | 416 | 0.81 |

Non-synonymous substitutions are highlighted in bold.
[a]Genomic position according to the reference sequence NC_004162.2.
[b]Amino acid changes.
[c]Amino acid position in the respective peptide.
[d]Frequency of the substitution in the dataset from the respective clade.

L210Q, I211T, K233E, K252Q) adaptative mutations have not been detected in the Brazilian isolates.

In an alignment analysis of all 422 new sequences, we found that among the total of CHIKV genome mutated positions the majority consisted of type C > T transitions (29.5%, $n = 368$) followed by T > C substitutions (22.9%, $n = 285$) (Fig. 3d). These transitions along with A > G and G > A transitions displayed a comparable frequency distribution across the sequence dataset, where mutations at several positions were identified in all new sequences (Fig. 3e). Due to the possibility of these substitutions resulting from sequencing errors, we performed the same comparative analysis on a different dataset containing sequences that form a subclade (from the South region) in clade I and that were generated by a different sequencing technology (Illumina) and a different research group (data in Supplementary files). We observed the same pattern of increased frequency (26.8%) of C > T transitions among those southern sequences, which suggested the observed increased transition changes in clades I and II were likely derived from mutation rather than sequencing artifacts.

## Discussion

Advances in sequencing technology and bioinformatics tools have contributed to an increased understanding of the genetic diversity and

transmission dynamics of emerging viruses causing epidemics[28]. In this study, we increased the number of CHIKV genomes deposited in the NCBI by generating 422 genomes that cover 12 Brazilian states over the years 2021–2022. By combining genomic and demographic data this effort has provided a comprehensive overview of CHIKV-ECSA phylogeny in Brazil which summarizes and updates previous phylogenies from other studies of localized outbreaks[10,12,17,29,30]. The updated and metadata-enriched dataset allowed us not only to describe the genetic diversity of circulating CHIKV variants but also to describe branching patterns observed in the updated phylogeny and infer the divergence time and likely location of emerging distinct subclades from the Brazilian lineage.

Following the arrival of ECSA lineage in the Northeast region, a number of other studies have predicted and reported the establishment and spread of this lineage to the rest of the country[10–12,17,20,29–32]. The Bayesian phylogeographic approach employed in this study has revealed the main routes of dispersion of CHIKV in the country. After being introduced in the Northeast region, the ECSA lineage dispersed towards several states from all five regions in Brazil. In line with previous findings[10,17,29,33–35], both the Bayesian phylogeographic and transmission network approaches indicated the Northeast region has been acting as the leading hub of virus spread towards other regions in the country, including by forming an intense virus exchange network with the Southeast region. These interactions are likely driven by human mobility between these areas, as both Southeast and Northeast regions are densely populated, housing transportation infrastructure and major urban centers that attract a substantial number of visitors. From the Southeast, the virus dispersed to the southern state of Paraná (posterior probability of 0.96) with the time of the most recent common ancestor shared with samples from São Paulo estimated to be January 2020. In addition to spreading in Brazil, CHIKV has also extended its circulation to other countries such as Paraguay and Haiti. CHIKV circulation in Paraguay and Haiti has been previously associated with transmission events originating from Brazil through international viral exchange likely mediated by human movement[36–38]. These results evidence the lineage's potential expansion across Latin America, despite limited information available about this lineage in the region as evidenced by the lack of CHIKV-ECSA sequences from other Latin American countries.

Our time-measured phylogeny corroborated previous studies and revealed emerging branching patterns, mainly represented by two well-supported distinct subclades named clade I and II. Both these clades were estimated to emerge in the Southeast region around the first months of 2018. At the end of that year, the Southeast region accounted for more than 65 thousand reported cases. Moreover, differences in sequence composition were observed between these clades, with clade I being more diverse as it comprises sequences from four different regions (Northeast, Midwest, Southeast, and South) collected in the years 2021–2022, while clade II contains mostly sequences from northeastern states collected in 2022. This difference in the clades sequence composition profile might change as the lineage continues to expand into the country. The observed differences in geographic diversity between clades might reflect distinct transmission networks underlying the divergence and expansion of these subclades.

Different viral lineages might be under distinct evolutionary pressure that together with the emergence and selection of mutations can drive viral adaptation to a particular environment[39,40]. It has been argued that two different CHIKV lineages, IOL and Asian, have undergone different evolutionary trajectories leading to different vector adaptive potentials[41]. Here, we used the ratio of non-synonymous (dN) to synonymous (dS) nucleotide substitutions in the CHIKV envelope gene to assess the selective regime to which clades I and II might be subjected. Our analysis revealed that both clades have experienced positive diversifying selection. The CHIKV *envelope* gene encodes three viral glycoproteins associated with membrane fusion

and receptor binding during infection, and these proteins are a target of neutralizing antibodies[42,43]. This gene constitutes a variable region in the virus genome where several adaptive mutations have been identified[44], for instance, mutations in the envelope proteins such as E1-226V and E2-L210Q have been implicated in the increased adaptation and transmission of the CHIKV IOL lineage in *Aedes albopictus*[22,45], leading to the epidemics reported in the Indian Ocean region between 2004 and 2007[46,47].

A higher ratio of non-synonymous substitutions is observed under a positive selection regime promoted by virus-host interactions[48]. Since SNVs continue to arise in RNA virus populations mainly driven by errors made by the virus replication complex that lead to genetic diversity, we compared the mutational profiles of clades I and II[49]. We identified in both clades isolates harboring mutations (E1-T98A and E2-V264A) previously associated with increased fitness in mosquitoes *Aedes ssp.* when these mutations are present in the background with either E1-226A[26] or E1-226V[27] mutations, although the latter was not observed in the genomes from this study. We identified several SNVs across nonstructural and structural protein genes that were exclusive to each clade. Clade II presented more SNVs ($n = 27$) than clade I ($n = 13$), of which three are non-synonymous substitutions (E1-T288I; nsP2-P352A, and nsP4-A43V). Literature research revealed that E1-T288I change was previously identified in a 2017 sequence from Iran and also in CHIKV sequences collected in 2016 from infected cancer patients in Rio de Janeiro, Brazil[50,51]. The nsP2-P352A substitution was also present in sequences collected between 2016 and 2017 in Rio de Janeiro[17,52]. In turn, clade I sequences contain six non-synonymous mutations across nonstructural (nsP2 and nsP4) and structural protein sequences (capsid, 6k, and E2). The nsP4-V555I change was previously detected in sequences from Thailand in 2008–2009[53]. The E2-L248F from clade I has been reported in Asian lineage sequences from Colombia (2014–2015) and Philippines (2012)[54,55]. Isolates from Thailand, Indonesia, Lao PDR, Cameroon and India also presented the 6K-I54V mutation observed in clade I[53,56–59]. Despite the detection of these mutations in different countries (indicating homoplasy) by other previous studies, there is no information about the functional impact of such substitutions on CHIKV fitness, thus warranting further experimental studies to elucidate the potential effects of SNVs on lineage-specific evolutionary adaptation. It has been argued that not only non-synonymous mutations have the potential to promote adaptive changes but also synonymous mutations, including deletions and insertions in the 3'untranslated region[60,61], can lead to changes in the viral RNA that can drive differential viral gene expression and host adaptation[60–62].

Mutational analysis of the 422 sequences generated in this study revealed a higher amount of C > T and T > C transitions followed by A > G and G > A substitutions in the CHIKV genome and several genome positions presenting these transitions with higher frequency across all new sequences. Moreover, Clade II has a significantly higher median frequency of C > T, A > G, and T > C transitions compared to clade I. Although this study cannot experimentally establish the significance of these transitions for CHIKV evolutionary adaptation, other studies have associated this mutational pattern with the action of host antiviral immune response mediated by AID/APOBEC and ADAR families of deaminases[63,64]. These enzymes are part of the interferon-stimulated innate immune response and promote viral genome transitions mutations by catalyzing the deamination of adenosine to inosine to cause A > G/T > C (by ADAR) substitutions or deamination of cytosine to uracil that leads to C > T/G > A (by AID/APOBEC) mutations[65]. This RNA editing process has been experimentally observed targeting specific viral sequences of SARS-CoV-2 to produce C > T transitions and increasing viral replication in Caco-2 cells, thus promoting viral increased fitness and adaptive evolution[66]. However, specific information about the effect of these RNA editing processes on the CHIKV genome remains elusive, although the *APOBEC3A* gene

has been observed up-regulated in the expression profile of CHIKV-infected patients[67]. Despite arguments that these transitions happen mainly in phylogenetically uninformative sites, the available evidence indicates that RNA editing processes might act as a significant driver of viral sequence diversity and evolutionary adaptation through the introduction of nucleotide changes[64,68].

This study recognizes that some limitations should be noted. Although this study presented the results of a Bayesian phylogeographic and mutational profile analysis performed on 422 new CHIKV sequences collected from 12 Brazilian states, not all states were evenly represented in the dataset used, which might limit the estimates relative to divergence time and ancestral location reconstructions, prompting careful interpretation of the results presented here. Ongoing sequencing efforts across the country could reduce this disparity in the future. Moreover, although single nucleotide substitutions identified among the new sequences offer insights into the evolutionary dynamics of CHIKV in Brazil, further functional studies need to be undertaken to elucidate the actual adaptive effect of these mutations.

Recurring CHIKF epidemics, as indicated by the seasonal peak patterns displayed in the incidence time-series plot, are evidence that the virus is endemic in Brazil. Human mobility, population immunity, vector suitability, vegetation coverage, site socioeconomic status, and viral sequence variation are factors considered to mediate the dispersal of CHIKV in Brazil[29,69–71]. Although we did not find the E1:226V *Aedes albopictus*-adaptive mutation in the Brazilian sequences, the high abundance in the region of widely spread competent vectors, such as *Ae. aegypti* and *Ae. albopictus*, together with favorable climatic and social conditions in large urban centers create conditions that modify the adaptive landscape of CHIKV, which in turn can allow the continued expansion of the ECSA lineage in the country with a resultant increased impact on public health[23,72,73]. Therefore, public health measures should be undertaken to ensure continuous genomic surveillance of circulating CHIKV variants which can help to identify viral transmission routes where focused vector control strategies could be employed to reduce the risk of recurring CHIKF epidemics.

## Methods

### Ethical statement
The project was approved by the Pan American World Health Organization (PAHO) and the Ministry of Health of Brazil (MoH) as part of the arboviral genomic surveillance efforts within the terms of Resolution 510/2016 of CONEP (Comissão Nacional de Ética em Pesquisa, Ministério da Saúde; National Ethical Committee for Research, Ministry of Health). This authorizes the use of clinical samples collected in the Brazilian Central Public Health Laboratories to accelerate knowledge building and contribute to surveillance and outbreak response. The study protocol, including collection and publication of individual-level data, was reviewed and approved by the Research Ethics Committee of the Universidade Federal de Minas Gerais with approval No. 32912820.6.1001.5149. Personally identifiable information was de-identified in the datasets and tables in a way that minimizes the risk of unintended disclosure of identity of individuals and information about them.

### Sample collection
Residual samples (serum or plasma) were obtained from the epidemiological surveillance routine of the Brazilian Central Public Health Laboratories (LACEN) from different states (Alagoas, Bahia, Goiás, Paraíba, Paraná, Pernambuco, Piauí, Maranhão, Minas Gerais, Mato Grosso do Sul, Rio Grande do Norte, and Sergipe). These samples, which were taken from patients that spontaneously seek medical care during epidemic season, were submitted to nucleic acid purification using the MagMax Viral RNA Mini kit (Thermo Fisher Scientific), following the manufacturer's recommendations, and were previously screened by each LACEN. CHIKV RT-qPCR positive

samples were selected for sequencing based on the cycle threshold (Ct) value ≤ 30 and the availability of demographic metadata such as sex, age, and municipality of residency. These demographic patient data were provided by LACENs and were collected through a questionnaire filled out by patients and/or health professionals at local health care services.

### cDNA synthesis and whole-genome sequencing
Extracted RNA from positive CHIKV samples were provided by collaborating LACENs and submitted to cDNA synthesis and PCR, using a sequencing protocol (primers sequences in Supplementary Data 1) based on multiplex PCR-tiling amplicon approach design for MinION nanopore sequencing[74]. All reactions were performed at biosafety level 2 facilities and using no template controls. PCR products were purified using 1x AMPure beads Beckman Coulter, UK) and quantified using Qubit 3.0 instrument (Life Technologies) and the Qubit dsDNA High Sensitivity assay. DNA library preparation was performed on all amplified samples using the Ligation Sequencing Kit (Oxford Nanopore Technologies). Individual samples were barcoded using the Native Barcoding Kit (NBD104, Oxford Nanopore Technologies, Oxford, UK). Sequencing library was loaded onto a R9.4 flow cell and data were collected for up to 48 sequencing hours.

### Generation of consensus sequences
Basecalling of raw FAST5 files and demultiplexing of barcodes were performed using the software Guppy v6.0.1 (https://github.com/nanoporetech). Consensus sequences were generated by a hybrid assembling approach implemented on Genome Detective (https://www.genomedetective.com/)[75].

### Phylogenetic reconstruction
We used MAFFT v.7 to align 422 new sequences (with coverage over >60% according to Thézé et al.[76], samples 736.22_RED, FS0116, and FS0132 were discarded) in addition to 1565 CHIKV whole-genome sequences publicly available in NCBI up to August 2022, forming a global dataset (n = 1987) that includes CHIKV all lineages. We used NCBI Entrez Utilities to retrieve worldwide CHIKV genomes according to the following inclusion criteria: chikungunya virus[title] AND 8000[SLEN]:13000[SLEN] for minimum (60% genome coverage) and maximum sequence length. Alignment of the global dataset can be found on the repository https://doi.org/10.6084/m9.figshare.22335331.v2.

This global dataset was used to infer a Maximum Likelihood (ML) phylogeny using the IQ-TREE 2.1.1 software[77,78]. Statistical support for tree nodes was estimated using the ultrafast bootstrap (UFBoot) feature implemented in IQ-TREE with 1000 replicates. We then used the ML tree from the global dataset to extract the Brazilian ECSA clade and use it to form a second dataset (total n = 713, 706 Brazilian sequences, 2 from Haiti, and 5 from Paraguay; sequences with genome coverage >60% according to Thézé et al.[76]) which was used to infer a time-scaled phylogeny using BEAST v1.10.4. First, we investigated the temporal signal regressing root-to-tip genetic distances from this ML tree against sample collection dates using TempEst v.1.5.1[79]. Secondly, we employed a stringent model selection analysis using both path-sampling (PS) and stepping-stone (SS) procedures to estimate the most appropriate molecular clock model for the Bayesian phylogenetic analysis[80]. For the Bayesian analysis, the uncorrelated relaxed molecular clock was chosen as indicated by estimating marginal likelihoods, also employing the HKY + G4 nucleotide substitution model, and the nonparametric Bayesian Skyline coalescent model. We combined two independent runs of 200 million states each[81]. The convergence of MCMC chains was checked using Tracer[82]. Maximum clade credibility (MCC) trees were summarized using TreeAnnotator v1.10.4 after discarding 10% as burn-in.

CHIKV ECSA lineage movements across Brazil were investigated using the Bayesian phylogeographic approach with a discrete trait

phylogenetic model. A trait file was used to discretize sequences sampling location by five Brazilian regions (North, Northeast, Southeast, South, and Midwest). For this analysis, we downsampled our Brazilian ECSA clade to a dataset containing 471 sequences to maximize the temporal signal in the dataset. The final downsampled dataset was assembled by removing closely related sequences from clades with repetitive collection dates and locations, thus avoiding bias in the analysis due to those oversampled clades.

MCMC analyses were performed in BEAST v1.10.4, running in duplicate for 200 million interactions and sampling every 20,000 steps in the chain. Convergence for each run was assessed in Tracer. MCC trees for each run were summarized using TreeAnnotator after discarding the initial 10% as burn-in. Finally, we used the SPREAD 4 tool to map spatiotemporal information embedded in the MCC trees[83].

### Transmission network analysis
A transmission network was reconstructed, using the StrainHub tool v1.1.2, from transition states summarized from the Bayesian phylogeography[84]. Centrality metrics on the tree nodes were also estimated for the network.

### Comparative mutational analysis
For comparative mutational analysis, we assembled separate alignment datasets for each subclade (clade I = 327; clade II = 168) and for all new sequences ($n = 422$). The sequence datasets were compared against the NCBI reference strain NC_004162.2 using MALVIRUS v1.0.2[85]. We filtered and selected substitutions with an occurrence frequency of 100% across the whole dataset and substitutions with a frequency above 60% across the envelope genes E1 and E2. Results were summarized in boxplots using Rstudio 2022.12.0.

### Selection pressure analysis
Since the ratio of non-synonymous (dN) to synonymous (dS) nucleotide substitutions can be used to study selection pressure on genomic sequences, we used the HYPHY software package that employs statistical methods that estimate the dN/dS to detect diversifying selection[86]. For that we performed an alignment of the envelope gene sequences of 720 Brazilian isolates and used BUSTED (restricting the analysis to each subclade I or II), an alignment-wide method implemented in HYPHY 2.5.42, that aims to detect evidence of episodic diversifying selection[87]. For comparison, we also used a different method, called MEME, a site-level approach, also implemented in HYPHY 2.5.42, that aims to detect evidence of both pervasive and episodic diversifying selection at individual sites (also restricting the analysis to each subclade I or II)[88].

### Epidemic curves from chikungunya fever cases reported in Brazil
Data of weekly notified suspected cases of chikungunya fever in Brazil from 2014 to 2022 (up to epidemiological week 28 of the year 2022) were supplied by the Ministry of Health of Brazil (MoH) (Ministério da Saúde, 2022). These data were used to calculate incidence and to plot time series charts using Rstudio 2022.12.0. The MoH defines a suspected case as all human cases presenting symptoms compatible with CHIKV infection (sudden onset of fever or intense arthritis not explained by other conditions), a case that resides or has traveled to endemic or epidemic areas up to 14 days prior to symptom onset, or that has an epidemiological link to an imported confirmed case. The MoH also informs that during epidemic season most cases are confirmed by clinical criteria only.

### Reporting summary
Further information on research design is available in the Nature Portfolio Reporting Summary linked to this article.

## Data availability
The new sequences generated in this study have been deposited in NCBI GenBank under accession numbers OQ759652- OQ760076 listed in Supplementary Data 1. Input data used for the phylogenetic and mutational profile analyses are provided on the repository https://doi.org/10.6084/m9.figshare.22335331.v2.

## Code availability
R codes used to create the charts are available on https://doi.org/10.6084/m9.figshare.22335331.v2.

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

## Acknowledgements

This work was supported by The Pan American Health Organization (PAHO/WHO), the Brazilian Ministry of Health grant SCON2021-00180 (Coordenação Geral de Laboratório de Saúde Pública-CGLAB and Coordenação Geral de Vigilância de Arboviroses-CGARB), the National Institutes of Health USA grant U01 AI151698 for the United World Arbovirus Research Network (UWARN). We thank the State Health Secretariats and the epidemiological surveillance services of Brazilian states and municipalities for all the support provided. J.X. was supported by Coordenação de Aperfeiçoamento de Pessoal de Nível Superior-Brasil (CAPES)-Finance Code 001. M.G. is funded by PON "Ricerca e Innovazione" 2014–2020. M.G. is supported in part by the CRP- ICGEB RESEARCH GRANT 2020 Project CRP/BRA20-03, Contract CRP/20/03. The authors would like to acknowledge the Global Consortium to Identify and Control Epidemics – CLIMADE (T.O., L.C.J.A., E.C.H., J.L., and M.G.) (https://climade.health/).

## Author contributions

Conception and design: L.C.J.A., M.G., V.F., and J.X.; investigations: J.X., M.G., L.C.J.A., V.F., A.M.B.D.F., Mau.L., E.C., Heg.F., C.O., N.G., Tal.A., M.E., E.S.R, E.V.S, D.D.L.R., L.D.M., S.T., A.N., A.R., A.F.M., A.L., A.V., A.L.S.D.M., B.V., C.A.M., C.Z., C.F., C.F.C.D.A, C.N.D.D.S, C.S.S., C.A.D.S., C.C.M.G., D.T., D.F.L.N., D.C., E.C.D.O., E.L.N.M., F.M.P., F.I., F.P.D.C., G.A., G.B., G.G.D.C.L., G.C.P., H.B., H.C.F.F., Hivy.F., I.G., I.N.R., I.R., I.C.D.S., J.S., J.M.R., J.L., J.A., J.P.M.D.N., J.N.W., J.P., J.J.F.D.M., K.G.L., L.G.L.N., L.C.V.F., L.B.D.S., L.S., L.A.F.D.S., L.A.P., L.D., M.C.B.C., M.G.A., M.A., Mar.L., M.C.S.U.Z., M.M.P., M.B.F., M.G.J., N.F., N.M., N.F.O.D.M., P.E.A.D.S., P.R., R.V.D.C., K.S.T., R.F.D.C.S., R.K., R.S., R.D.J., R.H.S., S.K., S.N.S., Tam.A., T.R., T.C., V.N., V.D.S., W.G.C., W.C.V.V., and W.N.A. Data curation: J.X., M.G., V.F., L.C.J.A; formal analysis: J.X., M.G., V.F., and L.C.J.A.; writing—original draft preparation: J.X., M.G., and L.C.J.A.; revision: J.X., M.G., V.F., and L.C.J.A.; resources: J.M.R., M.A., W.N.A., A.R., R.F.D.C.S., C.F.C.D.A, W.C.V.V., P.R., M.G.J., J.N.W., R.S., E.L.N.M., P.E.A.D.S., H.C.F.F., C.F., N.F.O.D.M., L.C.V.F., A.M.B.D.F., and L.C.J.A.

## Competing interests

The authors declare no competing interests.

## Additional information

Joilson Xavier [1,2,38], Luiz Carlos Junior Alcantara[1,2,38] ✉, Vagner Fonseca [3,38], Mauricio Lima[1,4], Emerson Castro[1,4], Hegger Fritsch [1,2], Carla Oliveira [5], Natalia Guimarães[4], Talita Adelino[4], Mariane Evaristo[6], Evandra S. Rodrigues[6], Elaine Vieira Santos[6], Debora de La-Roque[6], Laise de Moraes [7], Stephane Tosta[1,2], Adelino Neto[8], Alexander Rosewell[3], Ana Flavia Mendonça[9], Anderson Leite[10], Andreza Vasconcelos [11], Arabela L. Silva de Mello[12], Bergson Vasconcelos[13], Camila A. Montalbano[14], Camila Zanluca [15], Carla Freitas[16], Carlos F. C. de Albuquerque [3], Claudia Nunes Duarte dos Santos[15], Cleiton S. Santos[7], Cliomar Alves dos Santos[17], Crhistinne C. Maymone Gonçalves[18], Dalane Teixeira[13], Daniel F. L. Neto[16], Diego Cabral[11], Elaine C. de Oliveira[19], Ethel L. Noia Maciel[20], Felicidade Mota Pereira [12], Felipe Iani[4], Fernanda P. de Carvalho [11], Gabriela Andrade[12], Gabriela Bezerra[17], Gislene G. de Castro Lichs[21], Glauco Carvalho Pereira[4], Haline Barroso[13], Helena Cristina Ferreira Franz[16], Hivylla Ferreira [22], Iago Gomes[23], Irina N. Riediger[24], Isabela Rodrigues[17], Isadora C. de Siqueira [7], Jacilane Silva[11], Jairo Mendez Rico[25], Jaqueline Lima[12], Jayra Abrantes[23], Jean Phellipe M. do Nascimento [10], Judith N. Wasserheit[26], Julia Pastor[11], Jurandy J. F. de Magalhães[11,27], Kleber Giovanni Luz [28], Lidio G. Lima Neto[22], Livia C. V. Frutuoso [29], Luana Barbosa da Silva[19], Ludmila Sena[17], Luis Arthur F. de Sousa [22], Luiz Augusto Pereira [9], Luiz Demarchi[21], Magaly C. B. Câmara[23], Marcela G. Astete[12], Maria Almiron[25], Maricelia Lima[30], Marina C. S. Umaki Zardin[21], Mayra M. Presibella[24], Melissa B. Falcão[31], Michael Gale Jr. [32], Naishe Freire[11], Nelson Marques[24], Noely F. O. de Moura[29], Pedro E. Almeida Da Silva[20], Peter Rabinowitz [33], Rivaldo V. da Cunha[34], Karen S. Trinta[34], Rodrigo F. do Carmo Said[3], Rodrigo Kato [16], Rodrigo Stabeli[3], Ronaldo de Jesus[16], Roselene Hans Santos[11], Simone Kashima[6], Svetoslav N. Slavov[6,35], Tamires Andrade[13], Themis Rocha[23], Thiago Carneiro [13], Vanessa Nardy[12], Vinicius da Silva[9], Walterlene G. Carvalho[8], Wesley C. Van Voorhis[36], Wildo N. Araujo[25], Ana M. B. de Filippis [5] ✉ & Marta Giovanetti [1,2,37] ✉

[1]Instituto René Rachou, Fundação Oswaldo Cruz, Belo Horizonte, Brazil. [2]Instituto de Ciências Biológicas, Universidade Federal de Minas Gerais, Belo Horizonte, Brazil. [3]Organização Pan-Americana da Saúde, Organização Mundial da Saúde, Brasília, Brazil. [4]Laboratório Central de Saúde Pública de Minas Gerais, Fundação Ezequiel Dias, Belo Horizonte, Brazil. [5]Instituto Oswaldo Cruz, Fundação Oswaldo Cruz, Rio de Janeiro, Brazil. [6]Fundação Hemocentro de Ribeirão Preto, Ribeirão Preto, Brazil. [7]Instituto Gonçalo Moniz, Fundação Oswaldo Cruz, Salvador, Brazil. [8]Laboratório Central de Saúde Pública do Piaui, Piauí, Brazil. [9]Laboratório Central de Saúde Pública de Goias, Goiânia, Brazil. [10]Laboratório Central de Saúde Pública de Alagoas, Maceió, Brazil. [11]Laboratório Central de Saúde Pública de Pernambuco, Natal, Brazil. [12]Laboratório Central de Saúde Pública da Bahia, Salvador, Brazil. [13]Laboratório Central de Saúde Pública da Paraíba, João Pessoa, Brazil. [14]Universidade Federal de Mato Grosso do Sul, Campo Grande, Brazil. [15]Instituto Carlos Chagas, Fundação Oswaldo Cruz, Curitiba, Brazil. [16]Coordenação Geral dos Laboratórios de Saúde Pública, Ministério da Saúde, Brasília, Brazil. [17]Laboratório Central de Saúde Pública de Sergipe, Aracaju, Brazil. [18]Secretaria de Saúde do Estado do Mato Grosso do Sul, Campo Grande, Brazil. [19]Laboratório Central de Saúde Pública do Mato Grosso, Cuiabá, Brazil. [20]Secretaria de Vigilância em Saúde e Ambiente, Ministério da Saúde, Brasília, Brazil. [21]Laboratório Central de Saúde Pública do Mato Grosso do Sul, Campo Grande, Brazil. [22]Laboratório Central de Saúde Pública do Maranhão, São Luís, Brazil. [23]Laboratório Central de Saúde Pública do Rio Grande do Norte, Natal, Brazil. [24]Laboratório Central de Saúde Pública do Paraná, Paraná, Brazil. [25]Pan American Health Organization, Washington, USA. [26]Department of Global Health and Medicine, University of Washington, Washington, USA. [27]Universidade de Pernambuco, Serra Talhada, Brazil. [28]Universidade Federal do Rio Grande do Norte, Natal, Brazil. [29]Coordenação Geral das Arboviroses, Ministério da Saúde, Brasília, Brazil. [30]Universidade Estadual de Feira de Santana, Feira de Santana, Brazil. [31]Secretaria de Saúde de Feira de Santana, Feira de Santana, Brazil. [32]Department of Immunology, University of Washington, Washington, USA. [33]Department of Environmental and Occupational Health Sciences, University of Washington, Washington, USA. [34]Fundação Oswaldo Cruz, Instituto de Tecnologia em Imunobiológicos, Rio de Janeiro, Brazil. [35]Center for Research Development, CDC, Butantan Institute, São Paulo, Brazil. [36]Department of Medicine, University of Washington, Washington, USA. [37]Sciences and Technologies for Sustainable Development and One Health, University of Campus Bio-Medico, Rome, Italy. [38]These authors contributed equally: Joilson Xavier, Luiz Carlos Junior Alcantara, Vagner Fonseca. ✉e-mail: luiz.alcantara@ioc.fiocruz.br; ana.bispo@ioc.fiocruz.br; giovanetti.marta@gmail.com

