## [Peer Review File · Nature Communications]

Increased interregional virus exchange and nucleotide diversity outline the expansion of chikungunya virus in BrazilREVIEWER COMMENTS

Reviewer #1 (Remarks to the Author):

This paper by Xavier et al. analyzes 422 CHIKV genomic sequences from 12 Brazilian states collected from 2021-2022 as part of genomic surveillance when more than 312,000 Cases were reported. Brazil remains an important endemic country for CHIKV seven years after its introduction so continued surveillance is important, and this paper presents the most up-to-date phylogeny for the country. The 422 genomes were combined with 1,565 CHIKV genomes retrieved from GenBank and analyzed using Bayesian methods. The results inferred the emergence of two distinct ECSA subclades in Brazil, with the northeastern part of the country serving as a hub of virus spread. The authors speculate that an increased frequency of C>T transitions in the newer genomes suggests that innate host defense factors may be driving CHIKV evolution in Brazil. Overall, the paper is very impressive in the scale of the surveillance and sequencing and the resolution of patterns of spread. However, the overall novelty of the findings is limited and some of the conclusions are overstated without alternative explanations. The analyses appear to be properly done and the conclusions are robust aside from the speculative nature of the mutational details. The writing is generally clear but could benefit from some reorganization of the introduction and a clearer explanation of how this major effort improves our mechanistic understanding of CHIKV epidemiology and spread. For example, are there clear differences between clades I and I in pathogenicity, vector infection or usage, transmission efficiency, etc.? The only information presented is on different viremia levels but it is not clear whether times between disease onset and sample collection were consistent between the clades.

Specific comments:

1. More information on the surveillance efforts in Brazil since 2014 and the incidence data presented would be useful; has the program been consistent, including clinical and lab criteria applied to identifying cases?
2. Virus names are capitalized only when they are formal places: e.g. chikungunya and dengue are not capitalized
3. Line 65: dispersal
4. Line 93: Reference 14 does not describe a fatal CHIKV case, and "commodities" should be comorbidities?.
5. Line 107: this sentence is misleading: "Since available vaccine candidates are based on non-Brazilian variants, increasing the availability of genomic data to characterize the genetic pool of the viral population circulating in Brazil might facilitate the future development of an efficient and more representative CHIKV vaccine." Experimental and epidemiologic studies have demonstrated that, regardless of the infecting genotype or vaccine strain, cross-protection against all other strains is robust.
6. Line 130: Were the 312,000 cases laboratory-confirmed?
7. Line 148: What was the depth of sequence coverage?
8. Line 153: How was the combination of partial and complete genomes analyzed?
9. Line 244: The differences in the viral loads between the two clades could reflect differences in pathogenicity, but also could be influenced by other factors such as differences in sample collection times relative to symptoms onset. Other potential explanations like this should be added.
10. Line 206: Several previous studies have described northeastern Brazil, the most affected region during the past 9 years, as a major hub of regional spread. These should be cited here or in the discussion.
11. Line 261 or discussion: Is there a spatial or other relationship between the T288I substitution in the E1 protein gene and the many E1 and E2 mutations shown to allow CHIKV to be transmitted more efficiently by mosquito vectors?
12. Line 306: Any hypotheses for the factors driving the patterns of geographic spread revealed in the analyses presented?
13. Line 310: Were clades I and I described previously?
14. Line 327: the statement "evolutionary pressure might be driven by the host antiviral immune response, as the viral envelope protein is a target for neutralizing antibodies" is not consistent with the strong consensus view that CHIKV infection generates long lived protection against reinfection, i.e. no adaptive immune selection on its evolution.
15. Line 352: Also the 3'UTR of CHIKV has been shown in several studies to have a strong impact

on several phenotypes.

16. Line 375: The association between socioeconomic factors and arbovirus transmission is still disputed. Please see Power et al. Socioeconomic risk markers of arthropod-borne virus (arbovirus) infections: a systematic literature review and meta-analysis. *BMJ Glob Health*. 2022 Apr;7(4):e007735. PMID: 35428678

17. Line 377: What about other E2 and E1 vector-adaptive mutations?

18. Line 570: How is "notified" different than "laboratory-confirmed?" CHIKV infection can easily be confused with dengue in the absence of laboratory diagnostics, so how certain are the "notified" cases?

Reviewer #2 (Remarks to the Author):

This work by Xavier et al. addresses an important and relevant question relating to the spatial-temporal spread and potentially significant mutations of Chikungunya virus that have occurred in Brazil. By working with local partners, the authors have significantly increased genomic sequencing capacity for Chikungunya virus, more than doubling the number of genomic sequences available for Brazil. By applying phylogenetic methods, they find that the North-east of Brazil is a primary source of viral expansion towards other regions. Moreover, they have identified several sites within the viral genome under diversifying selection which future experimental work could build upon. Overall, I found the manuscript to be well-written and informative. However, I do have some concerns that I believe should be addressed before publication.

I suggest that the authors provide a plot showing the number of genomic sequences by the number of cases to help the reader understand the level of reporting bias and if there are any significant reporting differences between states/regions.

Additionally, in Figure 2B, the authors show the number of genomes with over 70% coverage, but within their study, they used sequences with over 60% coverage. I suggest that the authors make this consistent throughout the manuscript.

Furthermore, I would like the authors to provide more information about how the global dataset was built. Specifically, I would appreciate a description of the inclusion/exclusion criteria. This information would help the reader interpret the number of viral introductions into Brazil and any possible biases that could affect the phylogenetic analyses.

The authors reduced their ECSA Brazilian dataset by ~33% to maximise temporal signal. However, the long branches to the North-Eastern regions in Figure 2A implies that there are potential unsampled locations which could affect the results of the phylogeography. Coupled with the further ~33% reduction in the dataset, if this was not done in a systematic and appropriate fashion it could further bias the phylogeography I.e. if some regions/states were over sampled compared to others. I would like the authors to explain in further detail how they downsampled their Brazilian ECSA dataset, specifically show that the downsampling was done so no spatiotemporal biases were introduced in the dataset, and show the distribution of sequences per region of the updated ECSA dataset.

In Figure 1B, the use of cases per 100,000 people shows a measure of incidence. However, raw case counts can be informative to observe viral transmission trends. Could figure 1B be modified to show the raw case counts. In addition to this, regional trends can be and should be described from the epidemiological perspective as well as from the phylogeographic analyses. To this point, in line 234, the authors mention that the seasonal epidemic pattern occurs where incidence peaks within the first few months of the year. It appears that this is true for all states apart from the North East, which peaks around mid-year. Additionally, in 2022, epidemic synchrony occurs between all the states. I would like the authors to offer some hypotheses as to why in 2020-2021, the north-east epidemic started later than other regions and why this changed in 2022.

Line 234 - semester → months

I recommend considering the use of violin plots instead of box plots to better illustrate the spread and distribution of the data.

I noticed inconsistencies in the formatting of the references. Some references have a space between the word and the reference, while others do not. I strongly advise the authors to ensure consistency in the formatting of references throughout the manuscript.

Reviewer #1 (Remarks to the Author):

General comment: This paper by Xavier et al. analyzes 422 CHIKV genomic sequences from 12 Brazilian states collected from 2021-2022 as part of genomic surveillance when more than 312,000 cases were reported. Brazil remains an important endemic country for CHIKV seven years after its introduction so continued surveillance is important, and this paper presents the most up-to-date phylogeny for the country. The 422 genomes were combined with 1,565 CHIKV genomes retrieved from GenBank and analyzed using Bayesian methods. The results inferred the emergence of two distinct ECSA subclades in Brazil, with the northeastern part of the country serving as a hub of virus spread. The authors speculate that an increased frequency of C>T transitions in the newer genomes suggests that innate host defense factors may be driving CHIKV evolution in Brazil. Overall, the paper is very impressive in the scale of the surveillance and sequencing and the resolution of patterns of spread. However, the overall novelty of the findings is limited and some of the conclusions are overstated without alternative explanations. The analyses appear to be properly done and the conclusions are robust aside from the speculative nature of the mutational details. The writing is generally clear but could benefit from some reorganization of the introduction and a clearer explanation of how this major effort improves our mechanistic understanding of CHIKV epidemiology and spread. For example, are there clear differences between clades I and I in pathogenicity, vector infection or usage, transmission efficiency, etc.? The only information presented is on different viremia levels but it is not clear whether times between disease onset and sample collection were consistent between the clades.

Reply: We thank the reviewer for the positive comment. We greatly appreciate your valuable feedback and have carefully considered all of your suggestions. In response, we have made revisions to address each of the specific questions you've raised. See below our point-by-point summary.

Specific comments:

Comment 1: More information on the surveillance efforts in Brazil since 2014 and the incidence data presented would be useful; has the program been consistent, including clinical and lab criteria applied to identifying cases?

Reply: We appreciate the reviewer for raising this point. Since its first detection in mid-September 2014 in Brazil, the country has implemented extensive efforts to closely monitor the real-time evolution of the emerging CHIKV pathogen across the Americas. The Brazilian Ministry of Health, in collaboration with Central Public Health Laboratories located in each Brazilian state and local health departments, plays a crucial role in conducting active epidemiological surveillance for CHIKV.

The surveillance activities encompass various important aspects. Firstly, they involve monitoring the number of reported suspected cases, enabling the tracking of CHIKV incidence and its geographical distribution. Additionally, diagnostic tests are performed to confirm suspected cases, contributing to accurate identification and reporting of CHIKV infections. Moreover, outbreak areas are identified through systematic surveillance, facilitating targeted control measures to limit the spread of the virus.

Since 2016, our research team has been actively supporting the Public Health Laboratories working on behalf of the Brazilian Ministry of Health and the Pan-American Health Organization. Our collaborative efforts aim to enhance the knowledge about the genetic diversity of CHIKV circulating in the country. In this regard, CHIKV RT-qPCR positive samples meeting specific criteria, such as a cycle threshold (Ct) value ≤ 30 and the availability of demographic metadata (e.g., sex, age, municipality of residency), are selected for sequencing. We have summarized the chikungunya fever incidence data in figure 1b and made citations in the introduction section about the studies published since 2014.

Comment 2: Virus names are capitalized only when they are formal places: e.g. chikungunya and dengue are not capitalized

Reply: We appreciate the comment and we have made the changes.

Comment 3: Line 65: dispersal

Reply: We appreciate the comment and we have made the changes.

Comment 4: Line 93: Reference 14 does not describe a fatal CHIKF case, and “commodities” should be comorbidities?

Reply: We appreciate the comment and we have made the changes. Reference 14 of Souza et al. (2019) was used as it provides an alternative analysis of the genomic epidemiology of the CHIKV-ECSA lineage in Brazil. In this sense, we mentioned reference 14 to support our comment on line 93, although we agree with the reviewer regarding the specific location of the reference, and we have placed it right after the correlated sentence on now line 82.

Comment 5: Line 107: this sentence is misleading: “Since available vaccine candidates are based on non-Brazilian variants, increasing the availability of genomic data to characterize the genetic pool of the viral population circulating in Brazil might facilitate the future development of an efficient and more representative CHIKV vaccine.” Experimental and epidemiologic studies have demonstrated that, regardless of the infecting genotype or vaccine strain, cross-protection against all other strains is robust.

Reply: We appreciate the reviewer's comments raising this topic. Following the initial discussion in the paragraph regarding the limited availability of CHIKV sequences from Brazilian epidemics, the sentence on line 107 (now line 71) intended to bring attention to the efforts of increasing the amount of CHIKV genomic data from Brazil as an approach to provide updated information for future studies including vaccine studies. To avoid a misleading sentence, we followed the reviewer's suggestion and we have restructured the introductory paragraph in the revised manuscript.

Comment 6: Line 130: Were the 312,000 cases laboratory-confirmed?

Reply: The 312,000 chikungunya fever cases mentioned in line 130 refers to the number of suspected cases notified to the Ministry of Health of Brazil (BrMoH) and available at the moment of writing the manuscript. We were able to obtain chikungunya fever suspected cases number notified to the BrMoH by epidemiological week (up to epi week 28 of 2022). The MoH defines “*suspected case*” as all human cases presenting symptoms compatible with CHIKV infection (sudden onset of fever or intense arthritis not explained by other conditions), that resides or have traveled to endemic or epidemic areas up to 14 days prior to symptom onset, or that has an epidemiologic link to an imported confirmed case. The BrMoH also informs that during epidemic season most cases are confirmed by clinical criteria only.

In order to reduce the risk of misinterpretation we have edited the text (now line 116) to provide a more detailed explanation of case classification in the methods section as well.

Comment 7: Line 148: What was the depth of sequence coverage?

Reply: The average depth reported in our study was 3,290x (range 58 to 129,706). We have edited the text to add a sentence containing the average sequencing depth.

Comment 8: Line 153: How was the combination of partial and complete genomes analyzed?

Reply: The analysis performed in this study is primarily based on sequence similarity assessed by sequence alignment. Therefore, all sequences either partial or complete were aligned together in order to evaluate their similarity, and consequently to infer their evolutionary relationship. In addition, the analyzed dataset was assembled by selecting sequences with a genome coverage over 60% as a minimum to reconstruct a reliable phylogeny, according to Thézé et al., 2018 (doi:10.1016/j.chom.2018.04.017). This is the same standard approach employed in previous virus genomic epidemiology studies, such as the publication from Adelino et al. 2021 (Nature Communications, <https://doi.org/10.1038/s41467-021-22607-0>) which generated more than 200 partial and complete dengue virus genome sequences to study the transmission history of DENV 1-2 in Brazil.

Comment 9: Line 244: The differences in the viral loads between the two clades could reflect differences in pathogenicity, but also could be influenced by other factors such as differences in sample collection times relative to symptoms onset. Other potential explanations like this should be added.

Reply: We appreciate the reviewer's suggestion, and we agree that the difference between the median Ct values observed in clades I and II might have originated from inconsistencies in sample size and, alternatively, from differences in the times between sample collection and disease onset. These possible influencing factors were stated in the revised manuscript from line 242. We also calculated that the number of days between the onset of symptoms and the collection of specimens was 0-32 days for clade I while clade II presented a range of 0-34.

Comment 10: Line 206: Several previous studies have described northeastern Brazil, the most affected region during the past 9 years, as a major hub of regional spread. These should be cited here or in the discussion.

Reply: We have made the necessary changes in the discussion section.

Comment 11: Line 261 or discussion: Is there a spatial or other relationship between the T288I substitution in the E1 protein gene and the many E1 and E2 mutations shown to allow CHIKV to be transmitted more efficiently by mosquito vectors?

Reply: Although our study was able to identify synapomorphies with clade II, it did so with the intent to communicate features observed in this emergent clade, as well as in clade I. Moreover, the T288I substitution could possibly represent a marker SNV reflecting solely the common evolutionary trajectory shared by the viruses from clade II. We also mentioned in the discussion section that T288I have been previously identified in sequences from Iran and Brazil, however, these studies did not provide any more information on this substitution. Putative spatial or functional associations between T288I and other mutations found in the *envelope* gene could be assessed by structural or functional studies, therefore requiring further investigation.

Comment 12: Line 306: Any hypotheses for the factors driving the patterns of geographic spread revealed in the analyses presented?

Reply: We appreciate the reviewer for raising this comment. While the present article does not extensively evaluate the factors driving the patterns of CHIKV geographic spread, we can propose some hypotheses (summarized from line 305 in the manuscript) for the observed interregional spreading:

1. Human mobility and travel appear to be the most suitable factors to consider when evaluating patterns involving the large exchange of viruses. The movement of infected individuals, whether through international travel or domestic migration, can contribute

to the introduction and dissemination of viruses in new regions. In the case of CHIKV, the observed patterns of virus exchange between the Southeast and Northeast regions of Brazil, as well as the exchange via the Midwest, can be attributed to the significant human mobility between these areas. Both the Southeast and Northeast regions are densely populated and house major urban centers that attract a substantial number of visitors, including tourists and business travelers. Additionally, the Midwest serves as a crucial transportation hub connecting Brazil with its neighboring countries. The intense flow of people in these areas, coupled with the high concentration of flight hubs and popular tourist destinations, could facilitate the spread of viruses. Furthermore, the interconnectedness of transportation networks, including air travel, may enable the rapid dissemination of viruses across different regions and even international borders.

Additional hypotheses still need to be considered including :

- Climate and ecological suitability: CHIKV transmission is strongly influenced by temperature, humidity, and other climatic factors. Areas with suitable environmental conditions for the vector (*Aedes* mosquitoes) and optimal viral replication are more likely to experience higher transmission rates and subsequent spread.
- Vector abundance and distribution: The presence and density of *Aedes* mosquitoes play a crucial role in CHIKV transmission. Factors such as urbanization, deforestation, and changes in land use can affect mosquito habitats and influence vector populations, thereby impacting the geographic spread of CHIKV.
- Socioeconomic and environmental factors: Socioeconomic factors, including poverty, inadequate sanitation, and limited access to healthcare, can contribute to the persistence and amplification of CHIKV transmission within communities.
- Immunity levels and previous exposure: The level of immunity within a population can affect the transmission dynamics of CHIKV. Areas with a higher proportion of susceptible individuals, either due to low previous exposure or a lack of vaccination, may experience more rapid and extensive spread of the virus.

Further studies focusing on investigating these patterns could be pivotal in identifying areas that need to be prioritized for surveillance and designing effective preventive measures.

Comment 13: Line 310: Were clades I and II described previously?

Reply: To the best of our knowledge this is the first study describing the emergence of clades I and II in the Brazilian CHIKV phylogeny. Our study summarizes an intense surveillance effort to sequence CHIKV genomes from recent outbreaks (2021-2022) across several Brazilian states. The generation of new CHIKV sequences from recent cases (not provided by other studies yet) allowed us to infer the most up-to-date comprehensive phylogeny of the ECSA lineage in Brazil, revealing the emergence and expansion of two distinct subclades.

Comment 14: Line 327: the statement “evolutionary pressure might be driven by the host antiviral immune response, as the viral envelope protein is a target for neutralizing antibodies” is not consistent with the strong consensus view that CHIKV infection generates long lived protection against reinfection, i.e. no adaptive immune selection on its evolution.

Reply: We appreciate the issue raised by the reviewer. The aforementioned line was intended to highlight the impact of the envelope gene sequence variability on the virus diversity. By agreeing with the reviewer's comments we have made changes in the paragraph (from now line 335) that now reads: "The CHIKV envelope gene encodes the three viral glycoproteins associated with membrane fusion and receptor binding during infection, and these proteins are a target of neutralizing antibodies(Quiroz et al. 2019; Jin et al. 2015). This gene constitutes a variable region in the virus genome where several adaptive mutations have been identified(Bartholomeeusen et al. 2023), for instance, mutations in the envelope proteins such as E1-226V and E2-L210Q have been implicated in the increased adaptation and transmission of the CHIKV IOL lineage in *Aedes albopictus*(Tsetsarkin et al. 2014; Tsetsarkin et al. 2007), leading to the epidemics reported in the Indian Ocean region between 2004-2007(Tsetsarkin et al. 2011; Mavalankar et al. 2008)".

Comment 15:Line 352: Also the 3'UTR of CHIKV has been shown in several studies to have a strong impact on several phenotypes.

Reply: We appreciate the reviewer's comment and we have updated the sentence to include the highlighted subject (see line 366 in the revised manuscript).

Comment 16: Line 375: The association between socioeconomic factors and arbovirus transmission is still disputed. Please see Power et al. Socioeconomic risk markers of arthropod-borne virus (arbovirus) infections: a systematic literature review and meta-analysis. *BMJ Glob Health*. 2022 Apr;7(4):e007735. PMID: 35428678

Reply: We appreciate the reviewer's suggestion of a new reference, however, our study does not intend to dive into socioeconomic factors associated with CHIKV infection, although we decided to mention it in the discussion's last paragraph in order to provide context to the reader, as these factors are being actively researched regardless of any dispute. In this sense, we cited a 2021 modeling study (Freitas et al., <https://doi.org/10.1371/journal.pntd.0009537>) using CHIKF cases from the city of Rio de Janeiro that revealed that areas at high risk of CHIKV transmission presented increased temperature, low vegetation, and low socioeconomic status (measured by a sociodevelopment index); these results are also in line with the overall findings from the suggested, now also cited, 2022 systematic literature review.

Comment 17: Line 377: What about other E2 and E1 vector-adaptive mutations?

Reply: We have provided a clearer presentation of the mutation profile of the CHIKV sequences generated in this study. Alignment of sequences from clade I and II revealed the absence of other vector-adaptive mutations in E1 (K211E, E1-A226V) and E2 (D60G, R198Q, L210Q, I211T, K233E, K252Q). A new sentence has been added in the discussion section (from line 346) and a new paragraph has been added in the results section (from line 265) that now reads: "Moreover, both clades contain genomes harboring the E1-T98A mutation (121/327 sequences in clade I, and 31/167 sequences in clade II), while only clade II reported 113 sequences (67%) presenting another mutation, the V264A in the E2 gene. These two mutations have been associated with increased virus infectivity in *Aedes* ssp. when presented in distinct genetic backgrounds (E1-226V with E1-T98A or E1-226A with E2-V264A)(Agarwal et al. 2016; Tsetsarkin et al. 2011). However, the E1-226V mutation, as well as other E1 (K211E) and E2 (D60G, R198Q, L210Q, I211T, K233E, K252Q) adaptive mutations have not been detected in the Brazilian isolates".

Comment 18:Line 570: How is “notified” different than “laboratory-confirmed?” CHIKV infection can easily be confused with dengue in the absence of laboratory diagnostics, so how certain are the “notified” cases?

Reply: We appreciate the reviewer's comments raising this question. Indeed, a more detailed explanation of case classification is required, so we have edited the text in the methods section to reduce the risk of misinterpretations. See **Comment 6** for more details.

Reviewer #2 (Remarks to the Author):

This work by Xavier et al. addresses an important and relevant question relating to the spatial-temporal spread and potentially significant mutations of Chikungunya virus that have occurred in Brazil. By working with local partners, the authors have significantly increased genomic sequencing capacity for Chikungunya virus, more than doubling the number of genomic sequences available for Brazil. By applying phylogenetic methods, they find that the North-east of Brazil is a primary source of viral expansion towards other regions. Moreover, they have identified several sites within the viral genome under diversifying selection which future experimental work could build upon. Overall, I found the manuscript to be well-written and informative. However, I do have some concerns that I believe should be addressed before publication.

Comment 1: I suggest that the authors provide a plot showing the number of genomic sequences by the number of cases to help the reader understand the level of reporting bias and if there are any significant reporting differences between states/regions.

Reply: We appreciate the reviewer's suggestion. Although we had provided bar charts depicting the number of CHIKV genome sequences per state beside the map of Brazil in Figure 1a, we are now providing a supplementary figure (Supplementary Figure 1) containing the number of sequences plotted over the time series of case counts reported for each geographic region.

Comment 2: Additionally, in Figure 2B, the authors show the number of genomes with over 70% coverage, but within their study, they used sequences with over 60% coverage. I suggest that the authors make this consistent throughout the manuscript.

Reply: We appreciate the reviewer's comments raising this issue and we have revised the text and made the changes.

Comment 3.: Furthermore, I would like the authors to provide more information about how the global dataset was built. Specifically, I would appreciate a description of the inclusion/exclusion criteria. This information would help the reader interpret the number of viral introductions into Brazil and any possible biases that could affect the phylogenetic analyses.

Reply: We appreciate the reviewer's comment and we have added in the methods section a more detailed explanation about the global dataset assembling. The edited sentence now reads: “We used NCBI Entrez Utilities to retrieve worldwide CHIKV genomes according to the following inclusion criteria: chikungunya virus[title] AND 8000[SLEN] : 13000[SLEN] for minimum (60% genome coverage) and maximum sequence length. Alignment of the global dataset can be found on the repository 10.6084/m9.figshare.22335331.”.

Comment 4: The authors reduced their ECSA Brazilian dataset by ~33% to maximize temporal signal. However, the long branches to the North-Eastern regions in Figure 2A implies that there are potential unsampled locations which could affect the results of the phylogeography. Coupled with the further ~33% reduction in the dataset, if this was not done in a systematic and appropriate fashion it could further bias the phylogeography I.e. if some regions/states were over sampled compared to others. I

would like the authors to explain in further detail how they downsampled their Brazilian ECSA dataset, specifically show that the downsampling was done so no spatiotemporal biases were introduced in the dataset, and show the distribution of sequences per region of the updated ECSA dataset.

Reply: We appreciate the reviewer's comments raising this question. Indeed a few long branches referring to the most recent sequences collected in 2022 in the Northeast region might represent unsampled locations, which is expected since this is a surveillance study based on available samples taken from patients that spontaneously seek medical care and then later were provided by local surveillance laboratories during the epidemic season. In this sense, we have provided a statement on the study limitations that addresses the sampling issue as it's not feasible to sample all cases.

We agree with the reviewer that further details should be provided about the dataset downsampling procedure which has been added in the revised manuscript (now from line 493). The Brazilian ECSA dataset containing 706 Brazilian sequences available up to August 2022 (plus 2 from Haiti, and 5 from Paraguay) was used as a reference to be downsampled and then used with the Bayesian phylogeographic approach taking into account a balanced diversity of sequences considering sample location and collection date. The final downsampled dataset (containing 471 sequences) was assembled by removing closely related sequences from clades with repetitive collection dates and locations, thus avoiding bias in the analysis due to those more representative clades with more sequences. By comparing with the MCC tree inferred from the 713 dataset, we observed that the reduced 471 dataset inferred a phylogeny (see Supplementary Figure 3) with the same topology observed in the MCC tree displayed in Figure 2a, suggesting no significant change occurred in the diversity and resemblance of the dataset after downsampling. As mentioned before, unequal sequencing efforts have been employed in different Brazilian states which consequently leads to some states being more represented than others, such as the case of the Northeast region that makes up 68% (8 states) of the reduced dataset (14% from Southeast, 3% from South, 4% from North, and 11% from Midwest), reflecting that this region accounts for the majority of cases reported in Brazil as shown in the time-series incidence data displayed in figure 1b .

Comment 5: In Figure 1B, the use of cases per 100,000 people shows a measure of incidence. However, raw case counts can be informative to observe viral transmission trends. Could figure 1B be modified to show the raw case counts. In addition to this, regional trends can be and should be described from the epidemiological perspective as well as from the phylogeographic analyses. To this point, in line 234, the authors mention that the seasonal epidemic pattern occurs where incidence peaks within the first few months of the year. It appears that this is true for all states apart from the North East, which peaks around mid-year. Additionally, in 2022, epidemic synchrony occurs between all the states. I would like the authors to offer some hypotheses as to why in 2020-2021, the north-east epidemic started later than other regions and why this changed in 2022.

Reply: We appreciate the reviewer. This is indeed an interesting point. We know that different factors might be associated with the resurgence of vector-borne infections. See below some hypotheses that may be able to explain why in 2020 and 2021 the Northeastern epidemic started later than other regions and why this has changed in 2022:

- Hypothesis 1: Climate and Environmental Factors

One possible hypothesis for the delayed onset of the CHIKV epidemic in the northeast region of Brazil in 2020-2021 could be attributed to climate and environmental factors. The northeastern region of Brazil is known for its distinct climate, characterized by high temperatures and high humidity. These climatic conditions may have created a less favourable

environment for the proliferation of the *Aedes aegypti* mosquito, the primary vector for transmitting CHIKV.

In 2022, there could have been changes in the climate patterns that made the northeastern region more conducive to the breeding and spread of mosquitoes. Factors such as temperature, rainfall patterns, and humidity levels might have become more favorable for mosquito reproduction and survival. These changes could have resulted in an increased mosquito population and subsequently led to the outbreak of the CHIKV epidemic in the region.

- Hypothesis 2: Population Susceptibility and Immunity

Another hypothesis could be related to population susceptibility and immunity. It is possible that the northeastern region had a higher level of pre-existing immunity to CHIKV compared to other regions in Brazil. If a significant portion of the population had been previously exposed to CHIKV or had developed immunity due to past outbreaks, it could have delayed the onset of the epidemic in 2020-2021.

However, over time, immunity levels may have waned or new individuals who were not previously exposed to the virus moved into the region, leading to a decrease in overall immunity. This change in population immunity dynamics could have made the northeastern region more vulnerable to CHIKV in 2022, resulting in the outbreak.

- Hypothesis 3: Viral Introduction and Spread

The introduction and spread of the CHIKV virus could also play a role in the delayed onset of the epidemic in the northeastern region of Brazil in 2020-2021. It is possible that the virus was introduced into other regions of the country before reaching the northeast. Factors such as travel patterns, migration, or the movement of infected individuals could have contributed to the initial spread of the virus in other parts of Brazil.

In 2022, the virus might have been introduced or reintroduced into the northeastern region, possibly through travel or the movement of infected individuals. Once introduced, the virus could have found suitable conditions for transmission and subsequently spread rapidly, leading to the outbreak in that year.

- Hypothesis 4: Surveillance and Reporting Bias:

Another hypothesis related to the likely-delayed onset of the CHIKV epidemic in the northeast region of Brazil in 2020-2021 could be attributed to surveillance and reporting biases. It is important to note that during that time, an explosion in the number of SARS-CoV-2 cases and deaths started to be reported within the country. Thus, it is possible that the surveillance systems in the northeast region were not as robust or efficient compared to other regions, resulting in a delayed detection and reporting of CHIKV cases. This delay could have given the impression that the epidemic started later in the northeast region. In 2022, with the gradual decrease in the number of notified SARS-CoV-2 cases, improvements in surveillance and reporting systems might have been implemented, leading to earlier detection and reporting of cases, thus changing the perception of the epidemic's timing in the northeast.

Additionally, in response to the reviewer's comment, we also examined the raw case counts to analyze the epidemiological dynamics of CHIKV cases from 2014 to 2022. Interestingly, we found that the new figure depicting raw case counts exhibited the exact same pattern as the previous figure. However, we have chosen to retain and present the previous figure in our study. The reason for this decision is that the number of CHIKV cases was normalized by considering the population size of each Brazilian region. This normalization process was undertaken to

prevent any biases arising from the different sample sizes and to gain deeper insights into the true epidemiological dynamics of CHIKV. Normalizing the case counts allows for a fair comparison between regions, providing a more accurate understanding of the relative burden of CHIKV within each region. Consequently, this normalization approach facilitates the identification of regions with a higher impact of CHIKV per capita, enabling targeted interventions, resource allocation, and informed public health decision-making.

Comment 6: Line 234 - semester → months

Reply: We appreciate the comment and we have made the changes.

Comment 7: I recommend considering the use of violin plots instead of box plots to better illustrate the spread and distribution of the data.

Reply: We appreciate the reviewer's suggestion and we have added to Figure 3 a violin plot under the boxplot, thus enriching the available data for the reader.

Comment 8: I noticed inconsistencies in the formatting of the references. Some references have a space between the word and the reference, while others do not. I strongly advise the authors to ensure consistency in the formatting of references throughout the manuscript.

Reply: We appreciate the comment and we have made the changes.

REVIEWERS' COMMENTS

Reviewer #1 (Remarks to the Author):

The authors have adequately responded to my previous comments related to interpretation of the data and technical details.